# OpenReview forum: "Draining Your Account: A Stealthy Attack on API Billing in Multi-Agent Systems"
_ICLR.cc/2026/Conference — Submitted to ICLR 2026_

### Official Review · Reviewer_kP7E · 2025-10-29

**Soundness:** 2
**Presentation:** 2
**Contribution:** 2
**Rating:** 2
**Confidence:** 4

**Summary:**

This paper proposes an attack method on API billing. In summary, the method modifies the tool description and tool functionality to hijack the multi-agent control flow, repeating API calls and resulting in large bills.

**Strengths:**

- API billing is an important problem.
- The paper evaluates the attack on 4 real-world agent frameworks.

**Weaknesses:**

- The evaluation is weak. It includes only 3 tasks and cannot demonstrate the generalizability of this method.
- Are the API provider and the attacker the same? If the provider itself is malicious, it will lose its reputation; once discovered, no one will use that API anymore. If they want to do it, why not stealthily add some charges to the user's account instead? The MPD, TCI, ER, RR would be 0, and more effective than Phantom.
- The attacker must extract the control flow of the target agents, which is a very strong assumption. It's not very applicable in the real world.
- The attack effectiveness is limited by the agent's control-flow design. If the control flow design is good and has a retry limit, the system will automatically stop after a few retries.
- The key point of this method is how to direct orchestration toward A* and persuade it to make redundant API invocations via P, but I did not find a detailed description of how to do this. The description in Section 4.1 is unclear: it describes content_analysis and summary APIs, but those appear in different tasks according to Appendix B. Can the authors provide an example of the entire attack trace?

**Questions:**

- Do all sub-agents share a single MCP server?
- Just to confirm, what does * mean in Appendix B? Does it indicate parts modified by the attacker?
- Is Phantom an automated attack pipeline, or do you manually extract the AST and analyze the control flow for each task?
- How does this differ from traditional control-flow hijacking? My understanding is that it just changes the attack goal from a direct malicious action to repeating benign actions to accrue charges.

---

> ### Author Response · Authors · 2025-11-26
> **Response to Reviewer kP7E (1)**
>
> We thank the reviewer for the thorough review and constructive feedback regarding the generalizability and implementation details.
> In our response, we present each entry formatted as follows: "W" for weaknesses, "Q" for questions, and "A" for corresponding answers, while citing your original review. "Line xx" indicate the corresponding  presentation in the revised manuscript that the authors upload, and the difference with the original one is displayed in blue.
>
> For the assumption of control flow extraction you especially concern, we discuss them in Section "Threat Model" (line 186) and add Appendix B titled "Generalizability to Closed-source MAS" (line 813) to elaborate on them. Moreover, for the implementation details, we  add Section 4.4 titled "Phantom Implementation" (line 290) and Appendix A titled "Implementation Details" (line 747), as well as Appendix E titled "Attack Implementation in Evaluation" (line 942), respectively. We respectfully look forward to your review.
>
> **W1: Generalizability of Tasks**
>
> > The evaluation is weak. It includes only 3 tasks and cannot demonstrate the generalizability of this method.
>
> **A(W1)**: Given the early stage of these research fields, we evaluate on representative and classical MASs. Phantom's success in these MASs sufficiently demonstrates its generalizability. Specifically:
>
> - *Domain Coding (T1)* implements the MAS proposed in the classical literature, showcasing advantages over single-agent systems and fostering extensive research development since then.
>
> - *Report Generation (T2)* is derived from classical literature and deployed with an officially-released MCP server, reflecting scenarios where MCP deployment is needed and encouraged.
>
> - *Collaborative Research (T3)*  is shared by Anthropic in their technical blog, "How We Built Our Multi-Agent Research System", which exemplifies production-level MAS implementations.
>
> - *Closed-source MAS* are vulnerable, as detailed in Appendix B with the generalized implementation and effectiveness analysis (line 800).
>
> **W2: API Provider and Attacker**
>
> > Are the API provider and the attacker the same? If the provider itself is malicious, it will lose its reputation; once discovered, no one will use that API anymore. If they want to do it, why not stealthily add some charges to the user's account instead? The MPD, TCI, ER, RR would be 0, and more effective than Phantom.
>
> **A(W2)**: *Phantom* is a strategy of generating malicious MCP servers, which is uploaded to MCP communities or marketplaces to reach potential victims, and we demonstrate a greedy API provider is highly motivated to deploy it for bill depletion. We further illustrate the "reputation" concern, and argue that Phantom is only a feasible attack scope.
>
> - **Reputation Concern for API Provider.** We demonstrate that proving the API provider as the attacker  is inherently challenging.
>   - **Ambiguous Attacker Identification.** The attack is facilitated by a malicious MCP server in MCP marketplaces or communities; however, it is hard to justify that this malicious MCP server is developed and uploaded by the API provider. Despite their motivation, definitive proof of the relationship between the uploader and the API provider is challenging.
>   - **Unclear Responsibility Attribution.** An illustrative anecdote involves large overage bills from numerous SMS invocation by an app. In this incident, the SMS company refuses refunds, claiming they merely executed what the app instructed, while the app insists that charges from other apps are not their responsibility ([reddit.com](https://www.reddit.com/r/zapier/comments/17pc4g0)).
>
> - **One Feasible Attack Scope.** We believe API providers have numerous methods to achieve their objectives, and we even provide some anecdotes in Appendix J (line 1282). We believe this work contributes in unveiling a potential attack routes that could lead to significant outcomes, and systematically evaluating their effectiveness and stealthiness.

---

> ### Author Response · Authors · 2025-11-26
> **Response to Reviewer kP7E (2)**
>
> **W3: Control Flow Extraction Assumption**
>
> > The attacker must extract the control flow of the target agents, which is a very strong assumption. It's not very applicable in the real world.
>
> **A(W3)**: We respond from the perspectives of the objective of extraction and generalizability to closed-source MAS. Additionally, we discuss this in Section "Threat Model" (line 186) and add Appendix B titled "Generalizability to Closed-source MAS" (line 813) in the revised manuscript.
>
> * **Objective of Extraction.** The objective is to identify the categories and conditions of control nodes that activate each agent, so that we manipulate them to direct orchestration towards the target agent.
>
> * **Closed-source MAS Generalizability.** The attack approach can be generalized to closed-source MAS. We discuss two typical assumptions: *with and without intermediate results* given each MAS query, which is the user instruction in this case.
>
>   - **With Intermediate Results.** The control flow extraction is achievable and the methodology is *equivalent* to that used in open-source MAS.
>     - **Rationale.** As MAS outputs the orchestration of agents processing each user instruction through the execution, it is feasible to extract the control flow to determine the activation of each agent, thereby identifying each control node; further, the conditions in activating respective agents can be iteratively obtained through multiple queries.
>     - **Effectiveness.** The control flow extraction allows for targeted hijacking to repeatedly activate the target agent, leading the effectiveness *consistent* with the original evaluation.
>
>   - **Without Intermediate Results.** The control flow extraction is not feasible; however, the attack approach remains *partially effective regarding corrupting the target agent*.
>     - **Rationale.** For each activation of the target agent during normal MAS execution, it is achievable to excessively invoke  API$^*$ via `malicious_invoke_pattern`($n_1$) with each MCP call, and rpersuade it to re-execute the MCP calls via  `persuasion_prompt`[$n_2$].
>     - **Effectiveness.** Failure to extract the control flow *diminishes* the attack effectiveness, falling to 4-6 times of Resource Wastage (RW) compared to that of 8-19 times in the open-source scenario. The statistics is equivalent to the metric of Invocation Density (ID) in the evaluation, which calculates the increased invocation of target API for each activation of target agent on average. The detailed values are presented in Table 2 in Section "Ablation Study", with the superior performance achieving an average of 6.21 times more invocation on Task 2 (line 478).
>
> **W4: Retry Limit**
>
> > The attack effectiveness is limited by the agent's control-flow design. If the control flow design is good and has a retry limit, the system will automatically stop after a few retries.
>
> **A(W4)**: We illustrate the motivation of this work and present the evaluation results against "retry limits".
>
> - **Motivation.** We aim to unveil this threat and evaluate its stealthiness against prevalent MAS and prominent mitigation, both of which demonstrate significant limitations. We believe that upon the *awareness* of this threat, the mitigation strategy can be developed from multiple perspectives , including development frameworks, external detection, and internal system design, which are all presented in the evaluation. In fact, we are actively involved in a production tool targeting vulnerability detection of MCP servers, and the attack proposed in this paper is incorporated as a valuable red-teaming strategy.
> - **Effectiveness of Retry Limits.** We implement mitigation of setting empirical retry limit, which is referred to as "empirical" and "Emp." for short. Table 4 presents the effectiveness of such mitigation, achieving an average detection rate of only 0.3 in complex MAS scenarios including five agents. The attribution lies in the varying API calls of the same MAS in response to different user instructions, and low thresholds of retry limits significantly impairing MAS performance. (line 518)
> - **Attack Performance against Retry Limits.** Table 5 presents the attack performance against MAS deployed with such mitigation, achieving an ASR of 0.94 and Resource Wastage (RW) of 2.5-5.4 times, as indicated in Table 5 under the "Emp." row. The attack is feasible by executing the maximum allowed invocations during each MAS execution. (line 534)

---

> ### Author Response · Authors · 2025-11-26
> **Response to Reviewer kP7E (3)**
>
> **W5: Implementation Details**
>
> > The key point of this method is how to direct orchestration toward A* and persuade it to make redundant API invocations via P, but I did not find a detailed description of how to do this. The description in Section 4.1 is unclear: it describes content_analysis and summary APIs, but those appear in different tasks according to Appendix B. Can the authors provide an example of the entire attack trace?
>
> **A(W5)**: We add Section 4.4 to illustrate "Phantom Implementation", providing pseudocode for the unified deployment on arbitrary MCP servers (line 290). We also provide an illustrative example covering the entire attack trace and with code snippets in Appendix A (line 747), as well as detailed attack implementation in MASs in the evaluation in Appendix E (line 942). In summary:
>
> * **Overview**. *Phantom* modifies tool and prompt primitives into $\tilde{T}$ and $\tilde{P}$ within the MCP server of the target API* called by target agent A*. Specifically, $\tilde{T}$ includes functions `metric_cn_manipulation`[$m_1$]  and `malicious_invoke_pattern`($n_1$); and $\tilde{P}$ includes prompts  `response_cn_manipulation`[$m_2$],  `persuasion_prompt`[$n_2$] , and `skip_mode_prompt`. The parameters $m_1$, $n_1$, $m_2$, and $n_2$ denotes maximum execution of each respective operation is executed. These crafts are implemented based on these caps in each phase.
>
> - **Phase1: Targeted Hijack.** (1) Extract control flow and classify control nodes into agent profile, task completion metrics, and response from other agents. (2) Decide the hijacking route and manipulate metric-based control nodes using `metric_cn_manipulation`[$m_1$] and response-based control nodes via `response_cn_manipulation`[$m_2$]. (3) Corrupt A* to excessively invoke API* via `malicious_invoke_pattern`($n_1$) with each MCP call, and re-execute the MCP via  `persuasion_prompt`[$n_2$].
>
> * **Phase2: Utility Preservation.** (1) Engage other essential agents concurrently to leverage their functionalities (Parallel Mode). (2) Skip agents with simple functions and merge them to A*  via `skip_mode_prompt` (Skip Mode).
> * **Phase3:  Exception Bypass.** Adjust parameters $m_1$, $m_2$, $n_1$, and $n_2$ to avoid triggering local loop alerts and timeout exceptions and modify `malicious_invoke_pattern`($n_1$) to mimic normal invocation to prevent excessive resource access warnings.
>
>
>
> **Q1: Technical Details**
>
> > Do all sub-agents share a single MCP server?
>
> **A(Q1)**: No, each MCP server corresponds to a specific API access, and each agent only has access to those defined in its system prompts and contributing to its functionality.
>
>
>
> **Q2: Notation**
>
> > Just to confirm, what does * mean in Appendix B? Does it indicate parts modified by the attacker?
>
> **A(Q2)**: No. The asterisk (*) denotes the target agent and target API for maximal activation (line 178). Modifications made by the attacker are denoted by a tilde cap (~), which is the tool and prompt in the MCP server for the target agent accessing the target API.
>
>
>
> **Q3: Automated Pipeline**
>
> > Is Phantom an automated attack pipeline, or do you manually extract the AST and analyze the control flow for each task?
>
> **A(Q3)**: No, the control flow is manually extracted in the evaluation of this paper. We further illustrate the extraction objectives, a potential automated solution, and existing obstacles.
>
> - **Objective of Extraction.** The objective is to identify the categories and conditions of control nodes that activate each agent, so that we manipulate them to direct orchestration towards the target agent.
> - **Possible Automation Solution.** An LLM agent equipped with Retrieval-Augmented Generation (RAG) based on a case pool intuitively facilitates the automation of such extraction.
> - **Obstacles.** However, the lack of benchmarks hinders us to train and evaluate such automation mechanisms. We hope to conduct further research regarding automated and large-scale attack and defense as the community evolves.

---

> ### Author Response · Authors · 2025-11-26
> **Response to Reviewer kP7E (4)**
>
> **Q4: Differentiation with Control Flow Hijacking**
>
> > How does this differ from traditional control-flow hijacking? My understanding is that it just changes the attack goal from a direct malicious action to repeating benign actions to accrue charges.
>
> **A(Q4)**: We describe existing control-flow hijacking work in MAS and clarify their difference.
>
> - **Control-flow hijacking in MAS.** The work of [Triedman et al. (2025)](https://arxiv.org/abs/2503.12188v1) appears to be the only non-concurrent work that defines and implements control flow hijacking in MAS (please correct as if we are mistaken). This work is conducted on MAS involving an orchestrator and  two agents, focuses on three static topological interactions among them (Round Robin, Central Orchestrator, and Central Orchestrator with external data structures) to achieve confused deputy through metadata shared with the Orchestrator. Our approach manipulates the judgment of control nodes within a complex MAS workflow involving up to five agents, simultaneously achieving targeted hijacking, utility preservation, and exception bypass, which has never been studied in the prior work.
> - **Indirect Prompt Injection (IPI).** Our works differentiates from IPI given the unique primitives of MCP. An agent deployed with MCP interact with an API through three primitives: (1) the tool invoking the API, (2) the resource representing the retrieval, and (3) the prompts assembling such retrieval to interact with LLM.
>   - **Method-wise.** Beyond prompt manipulation, we modify the tool involving precise execution, which collaborates with prompts to manipulate both the agent that invoke the target API and other agents that serves as control nodes.
>   - **Baseline.** Prompt Infection (PI) and Breaking Agents (BA) as baseline attacks fundamentally represent IPI with different attack vectors. PI injects malicious prompts via returned content through external invocation, while BA compromises an agent to append malicious commands to its responses.
> - **Difference.** The distinct attack scope and objectives naturally lead to different strategies to craft the modifications. Our work simultaneously achieves targeted hijacking, utility preservation, and exception bypass, which have not been performed in prior work.

---

### Official Review · Reviewer_S3DX · 2025-10-29

**Soundness:** 2
**Presentation:** 1
**Contribution:** 1
**Rating:** 2
**Confidence:** 4

**Summary:**

The authors introduce an attack called Phantom which generates malicious MCP servers that target and compel agents in a multi-agent system to make redundant API calls. The attack is evaluated on three MAS tasks with multiple frameworks and models.

**Strengths:**

- The threat to repeated API usage is interesting and important.
- The evaluations and ablations are well-designed and scaled.

**Weaknesses:**

- **Differentiation from Prior Work:** There is a large body of pre-existing work on attacks via the the [MCP protocol](https://invariantlabs.ai/blog/mcp-security-notification-tool-poisoning-attacks), [control flow hijacks on multi-agent systems](https://arxiv.org/abs/2503.12188), and indirect prompt injections. Very little of this existing literature is cited and/or engaged with and it is not immediately clear how their work differentiates from techniques in any of this prior work (although the goals may be different).
- **Writing & Math:** The presentation is often confusing or lacking, making the paper hard to follow. Some (non-exhaustive) examples:
  - The authors mention that they "generat[e] malicious MCP servers," but never specify what exactly that entails. In the *threat model* formalism in Section 3, it looks like their intervention may have something to do with the tooling / prompts, but no details or examples are given. They also describe different "modes" of the attack without showing how the modes are implemented within a server.
  - Imprecise definitions of Control Flow, Control Nodes, and Orchestration.
  - Nonstandard usage of formalism: *e.g.,* summations over $Y_t$ (intermediate outputs), inconsistent usage of sub- and super-scripts, double definitions of $\mathcal{O}^t$, nonstandard usage of set notation.
- **Threat Model.** Unlike other previous work on MCP attacks (e.g., [invariant](https://invariantlabs.ai/blog/mcp-security-notification-tool-poisoning-attacks)), this attack requires knowledge of the specific open-source MAS that's used to call the server and the specific tasks / queries.
- **Baselines:** There exist significantly stronger baselines in both the MCP attack literature and the multi-agent system attack literature, that this attack should be compared against (examples above).

**Questions:**

1. How are the methods in this work distinct from previous IPI and MCP attacks?
2. What exactly does Phantom alter in an MCP server? Is it general to all MCP servers? What benign MCP servers did you test on?

---

> ### Author Response · Authors · 2025-11-26
> **Response to Reviewer S3DX (1)**
>
> We thank the reviewer for the thorough review and constructive feedback regarding related work, implementation details, threat model, and presentation.
> In our response, we present each entry formatted as follows: "W" for weaknesses, "Q" for questions, and "A" for corresponding answers, while citing your original review. "Line xx" indicate the corresponding  presentation in the revised manuscript that the authors upload, and the difference with the original one is displayed in blue.
>
> For the implementation details that you especially concern, we  add Section 4.4 titled "Phantom Implementation" (line 290) and Appendix A titled "Implementation Details" (line 747), as well as Appendix E titled "Attack Implementation in Evaluation" (line 942), respectively. Moreover, for the generalizability to closed-source MAS, we discuss them in Section "Threat Model" (line 186) and add Appendix B titled "Generalizability to Closed-source MAS" (line 813) to elaborate on them.
> We respectfully look forward to your review.
>
>
>
> **W1.1& Q1: Differentiation from Prior Work**
>
> > How are the methods in this work distinct from previous IPI and MCP attacks?
> >
> > There is a large body of pre-existing work on attacks via the the [MCP protocol](https://invariantlabs.ai/blog/mcp-security-notification-tool-poisoning-attacks), [control flow hijacks on multi-agent systems](https://arxiv.org/abs/2503.12188), and indirect prompt injections.
>
> **A(W1.1& Q1)**: We have thoroughly reviewed the literature you mentioned, discussing the differentiation from the perspectives of MCP, control flow hijacking, and indirect prompt injections (IPI). We discuss the related work regarding MAS security and add those of MCP in Appendix K (line 1299), including comparisons with control flow hijacking methods you mentioned.
>
> - **MCP.** Existing works targeting MCP servers are conducted against single agents. Therefore, they lack the design involving control flow manipulation, showcasing limited impact of the MCP as an attack scope.
> - **Control Flow Hijacking in MAS.** The work of [Triedman et al. (2025)](https://arxiv.org/abs/2503.12188v1) appears to be the only non-concurrent work that defines and implements control flow hijacking in MAS (please correct as if we are mistaken). This work is conducted on MAS involving an orchestrator and two agents, focusing on three static topological interactions among them (Round Robin, Central Orchestrator, and Central Orchestrator with external data structures). It achieves confused deputy through metadata shared with the Orchestrator. Our approach manipulates the judgment of control nodes within a complex MAS workflow involving up to five agents, simultaneously achieving targeted hijacking, utility preservation, and exception bypass, which has never been studied in the prior work.
> - **IPI.** Our works differentiates from IPI given the unique primitives of MCP. An agent deployed with MCP interact with an API through three primitives: (1) the tool invoking the API, (2) the resource representing the retrieval, and (3) the prompts assembling such retrieval to interact with LLM.
>   - **Method-wise.** Beyond prompt manipulation, we modify the tool involving precise execution, which collaborates with prompts to manipulate both the agent that invoke the target API and other agents that serves as control nodes.
>   - **Baseline.** Prompt Infection (PI) and Breaking Agents (BA) as baseline attacks fundamentally represent IPI with different attack vectors. PI injects malicious prompts via returned content through external invocation, while BA compromises an agent to append malicious commands to its responses.
> - **Technical Differentiation.** The distinct attack scope and objectives naturally lead to different strategies to craft the modifications. Our work simultaneously achieves targeted hijacking, utility preservation, and exception bypass, which have not been performed in prior work.

---

> ### Author Response · Authors · 2025-11-26
> **Response to Reviewer S3DX (2)**
>
> **W2.1 & Q2: Implementation Details**
>
> > The authors mention that they "generat[e] malicious MCP servers," but never specify what exactly that entails. In the *threat model* formalism in Section 3, it looks like their intervention may have something to do with the tooling / prompts, but no details or examples are given. They also describe different "modes" of the attack without showing how the modes are implemented within a server.
> >
> > What exactly does Phantom alter in an MCP server? Is it general to all MCP servers?
>
> **A(W2.1& Q2)**: Phantom alters the tool and prompt primitives in an MCP server. Yes, it is general to all MCP servers.
> We add Section 4.4 to illustrate "Phantom Implementation", providing pseudocode for the unified deployment on arbitrary MCP servers (line 290). We also provide an illustrative example covering the whole attack trace and with code snippets in Appendix A (line 747), as well as detailed attack implementation in MASs in the evaluation in Appendix E (line 942). In summary:
>
> * **Overview**. *Phantom* modifies tool and prompt primitives into $\tilde{T}$ and $\tilde{P}$ within the MCP server of the target API* called by target agent A*. Specifically, $\tilde{T}$ includes functions `metric_cn_manipulation`[$m_1$]  and `malicious_invoke_pattern`($n_1$); and $\tilde{P}$ includes prompts  `response_cn_manipulation`[$m_2$],  `persuasion_prompt`[$n_2$] , and `skip_mode_prompt`. The parameters $m_1$, $n_1$, $m_2$, and $n_2$ denotes maximum execution of each respective operation is executed. These crafts are implemented based on these caps in each phase.
>
> - **Phase1: Targeted Hijack.** (1) Extract control flow and classify control nodes into agent profile, task completion metrics, and response from other agents. (2) Decide the hijacking route and manipulate metric-based control nodes using `metric_cn_manipulation`[$m_1$] and response-based control nodes via `response_cn_manipulation`[$m_2$]. (3) Corrupt A* to excessively invoke API* via `malicious_invoke_pattern`($n_1$) with each MCP call, and re-execute the MCP via  `persuasion_prompt`[$n_2$].
>
> * **Phase2: Utility Preservation.** (1) Engage other essential agents concurrently to leverage their functionalities (Parallel Mode). (2) Skip agents with simple functions and merge them to A*  via `skip_mode_prompt` (Skip Mode).
> * **Phase3:  Exception Bypass.** Adjust parameters $m_1$, $m_2$, $n_1$, and $n_2$ to avoid triggering local loop alerts and timeout exceptions and modify `malicious_invoke_pattern`($n_1$) to mimic normal invocation to prevent excessive resource access warnings.
>
> > What benign MCP servers did you test on?
>
> **A(W2.1& Q2)**: A benign MCP server refers to the original server that normally calls the target API. The process is as follows: benign MCP server (which calls the target API) → Phantom modifies its tools and prompts → malicious MCP server.
>
> **W2.2: Definition and Formalization**
>
> > Imprecise definitions of Control Flow, Control Nodes, and Orchestration.
> >
> > Nonstandard usage of formalism: *e.g.,* summations over Y_t (intermediate outputs), inconsistent usage of sub- and super-scripts, double definitions of O^t, nonstandard usage of set notation.
>
> **A(W2.2)**:
>
> - **Definitions.**
>
>   - Control Flow outlines the collaboration and coordination of agents in the MAS to facilitate the execution of user instructions.
>   - Control Node are determinants of agent activation within the control flow, and their states reflect intermediate outputs from previous execution. Control flow and states of control nodes collectively determine the orchestration.
>   - Orchestration denotes agents activated for continued execution.
> - **Formalization.** Summations over $Y_t$ denotes their accumulation over each step. Sorry that we do not observe double definitions of $O^t$; it denotes the orchestration at step t, indicating which agent(s) are activated to continue the MAS execution. As for the nonstandard use of set notation, does this refer to MCP\{T, R, P\}? This notation indicates that an MCP server is composed of these primitives, collectively forming an MCP server.

---

> ### Author Response · Authors · 2025-11-26
> **Response to Reviewer S3DX (3)**
>
> **W3: Threat Model**
>
> > **Threat Model.** Unlike other previous work on MCP attacks (e.g., [invariant](https://invariantlabs.ai/blog/mcp-security-notification-tool-poisoning-attacks)), this attack requires knowledge of the specific open-source MAS that's used to call the server and the specific tasks / queries.
>
> **A(W3)**: We respond from the perspectives of the generalizability to closed-source MAS, the representativity of tasks in evaluation, and a comparison with the invariant attack.
>
> - **Closed-source MAS Generalizability.** The attack approach can be generalized to closed-source MAS. We discuss two typical assumptions: *with and without intermediate results* given each MAS query, which is the user instruction in this case. Additionally, we discuss this in Section "Threat Model" (line 186) and add Appendix B titled "Generalizability to Closed-source MAS" (line 813) in the revised manuscript.
>   - **With Intermediate Results.** The control flow extraction is achievable and the methodology is *equivalent* to that used in open-source MAS.
>     - **Rationale.** As MAS outputs the orchestration of agents processing each user instruction through the execution, it is feasible to extract the control flow to determine the activation of each agent, thereby identifying each control node; further, the conditions in activating respective agents can be iteratively obtained through multiple queries.
>     - **Effectiveness.** The control flow extraction allows for targeted hijacking to repeatedly activate the target agent, leading the effectiveness *consistent* with the original evaluation.
>   - **Without Intermediate Results.** The control flow extraction is not feasible; however, the attack approach remains *partially effective regarding corrupting the target agent*.
>     - **Rationale.** For each activation of the target agent during normal MAS execution, it is achievable to excessively invoke  API$^*$ via `malicious_invoke_pattern`($n_1$) with each MCP call, and rpersuade it to re-execute the MCP calls via  `persuasion_prompt`[$n_2$].
>     - **Effectiveness.** Failure to extract the control flow *diminishes* the attack effectiveness, falling to 4-6 times of Resource Wastage (RW) compared to that of 8-19 times in the open-source scenario. The statistics is equivalent to the metric of Invocation Density (ID) in the evaluation, which calculates the increased invocation of target API for each activation of target agent on average. The detailed values are presented in Table 2 in Section "Ablation Study", with the superior performance achieving an average of 6.21 times more invocation on Task 2 (line 478).
> - **Evaluation Task Representativity.** Given the early stage of MAS and MCP research fields, we evaluate on representative and classical MASs. Phantom's success in these MASs sufficiently demonstrates its generalizability. Specifically:
>   - *Domain Coding (T1)* implements the MAS proposed in the classical literature, showcasing advantages over single-agent systems and fostering extensive research development since then.
>
>   - *Report Generation (T2)* is derived from classical literature and deployed with an officially-released MCP server, reflecting scenarios where MCP deployment is needed and encouraged.
>
>   - *Collaborative Research (T3)*  is shared by Anthropic in their technical blog, "How We Built Our Multi-Agent Research System", which exemplifies production-level MAS implementations.
>
>   - *Closed-source MAS* are vulnerable, as detailed in Appendix B with the generalized implementation and effectiveness analysis (line 800).
> - **Comparison with Invariant Attacks.** Invariant attacks target only single agents and do not incorporate the concept of control flow.

---

> ### Author Response · Authors · 2025-11-26
> **Response to Reviewer S3DX (4)**
>
> **W4: Stronger Baselines**
>
> > **Baselines:** There exist significantly stronger baselines in both the MCP attack literature and the multi-agent system attack literature, that this attack should be compared against (examples above).
>
> **A(W4)**: We elaborate on the choice of our baselines and argue that Breaking Agent (BA) is equivalent to the [control flow hijacking](https://arxiv.org/abs/2503.12188) attack, while being stronger than [invariant](https://invariantlabs.ai/blog/mcp-security-notification-tool-poisoning-attacks) attack you referenced.
>
> - **Design of Baselines.** We cover primary attack scopes, including user instructions, tool poisoning, and compromised agents. In terms of attack setting, we conduct these attacks with lower requirements, focusing solely on achieving maximal invocation, while employing more relaxed threat models adhering to their respective literature (line 388).
> - **BA Equivalent to Control Flow Hijacking Attack.** BA compromises an agent to definitively append the malicious commands to the end of its response, which is equivalent to polluting the shared "meta-data" communicated with the target agent in [control flow hijacks on multi-agent systems](https://arxiv.org/abs/2503.12188).
> - **BA Stronger than Invariant Attack.** [Invariant](https://invariantlabs.ai/blog/mcp-security-notification-tool-poisoning-attacks) attack manipulates the behavior of a single agent through tool poisoning, while these instructions may not be originally transmitted to the target agent passing through other agents. In contrast, BA definitively input the malicious instruction to the target agent.
> - **Invariant Attack vs. "Corruption" Process.** [Invariant](https://invariantlabs.ai/blog/mcp-security-notification-tool-poisoning-attacks) attack fails to hijack the control flow. Therefore, its maximum effectiveness is equivalent to the "corruption" of target agent in our attack, which only repeatedly invoke the target API when the target agent is naturally triggered. The performance is indicated by Invocation Density (ID) metric presented Table 2, leading to the resource wastage of 4-6 times.
>
> We would highly appreciate it if you could provide specific baseline methods you would like us to compare with ours.

---

### Official Review · Reviewer_VefN · 2025-10-31

**Soundness:** 3
**Presentation:** 3
**Contribution:** 3
**Rating:** 6
**Confidence:** 3

**Summary:**

This paper introduces Phantom, a framework for executing stealthy billing attacks against API usage in Multi-Agent Systems (MAS). This paper identifies a critical vulnerability in which a malicious Model Context Protocol (MCP) server can manipulate an MAS to surreptitiously inflate API calls, thereby draining a user's account. The proposed attack framework addresses three core challenges: (1) reliably activating a specific agent despite dynamic orchestration, (2) injecting numerous extraneous API calls without degrading task performance, and (3) evading built-in detection mechanisms. The evaluation demonstrates that Phantom can increase targeted API invocations by up to 26× with a 98% average success rate, while also successfully bypassing six distinct mitigation strategies with 94% effectiveness.

**Strengths:**

Originality: This paper identifies, formalizes, and addresses a novel and economically motivated threat vector in Multi-Agent Systems: covert API billing attacks. The paper clearly articulates the core challenges of such an attack and proposes a comprehensive framework to solve them.

Quality: The threat model is well defined, the attack framework is thoughtfully designed, and the claims are substantiated through extensive evaluation across multiple tasks and industrial frameworks.

Clarity: The paper logically structures the problem, the attack mechanics, and the experimental results. It provides a balanced perspective by thoroughly evaluating both the effectiveness of the attack and the limitations of potential mitigation strategies.

Significance: By demonstrating an extremely high attack success rate (98%) while showing that existing mitigation techniques are largely ineffective, the paper highlights a critical and timely vulnerability.

**Weaknesses:**

1. The evaluation of the baselines could be strengthened. First, the paper should explicitly state the total number of attack attempts conducted for each baseline. Second, the conclusion that the Breaking Agents (BA) baseline "demonstrates superior performance" is not rigorously supported by the data. The BA baseline is not the top performer across all scenarios (e.g., Task 1), and citing the maximum ASR (0.56) and maximum RW (2.51) from different tasks makes the claim appear overstated and less methodologically sound.

2. The choice of external mitigation tools could be more robust. The paper relies on general-purpose static analyzers (CodeQL, Codacy) that are not specialized for the unique logic of AI applications or MCP servers. A more compelling evaluation would involve testing against a static analyzer designed specifically for AI/LLM application security.

**Questions:**

1. The paper states that the targeted hijack phase requires extracting the control flow from the MAS source code. Does this imply that the control flow analysis must be manually conducted for each distinct MAS framework? If so, how would this attack approach generalize to closed-source MAS frameworks where the source code is unavailable? Would the attack's effectiveness (as measured by ASR, RW, etc.) be significantly diminished in such a black-box scenario?

2. The study focuses specifically on the Model Context Protocol (MCP). Has this paper investigated whether this attack vector is viable against Multi-Agent Systems that utilize other common agent-tool interaction protocols (e.g., direct RESTful API calls)? If so, were there significant differences in the results or the attack methodology? If not, could this paper speculate on how the choice of protocol might influence the attack's feasibility and success?

**Details Of Ethics Concerns:**

While the paper's topic is a valuable contribution to MAS security, the development of a functional attack framework necessitates a clear plan for responsible disclosure. This paper mentioned that the code repository is restricted to applicants who have passed a qualification review.

---

> ### Author Response · Authors · 2025-11-26
> **Response to Reviewer VefN (1)**
>
> We thank the reviewer for the thorough review and constructive feedback on generalizability discussions regarding closed-source MAS and broader attack vector.
> In our response, we present each entry formatted as follows: "W" for weaknesses, "Q" for questions, and "A" for corresponding answers, while citing your original review. "Line xx" indicate the corresponding  presentation in the revised manuscript that the authors upload, and the difference with the original one is displayed in blue.
>
> For the generalizability to closed-source MAS and its effectiveness that you especially concern, we discuss them in Section "Threat Model" (line 186) and add Appendix B titled "Generalizability to Closed-source MAS" (line 813) to elaborate on them. We respectfully look forward to your review.
>
> **W1.1: Baseline Evaluation (Attack Attempts of Baselines)**
>
> > First, the paper should explicitly state the total number of attack attempts conducted for each baseline.
>
> **A(W1.1)**: **120 attempts** are conducted for each baseline, which is comparable to the 100 attempts of our approach. We clarify this setting in the revised manuscript (line 392).
>
> * **Attempt Design.** We design 12 strategies from the perspectives of syntactic structure, the specification of the targeted agent, and the designation of the targeted API, following the SOTA work, and generates ten variants from each strategy.
>
>
>
> **W1.2: Baseline Evaluation (Sound Discussion)**
>
> > Second, the conclusion that the Breaking Agents (BA) baseline "demonstrates superior performance" is not rigorously supported by the data. The BA baseline is not the top performer across all scenarios (e.g., Task 1), and citing the maximum ASR (0.56) and maximum RW (2.51) from different tasks makes the claim appear overstated and less methodologically sound.
>
> **A(W1.2)**: We thank the reviewer for this constructive suggestion. We have made the following modifications, and detailed them in the revised manuscript (Line 504). "Overall, BA demonstrates superior performance on average across three tasks, with the best performing ASR of 0.56 on T2, and the highest RW of 2.51 on T3, both deployed with CAMEL framework."
>
> **W2: Robust External Mitigation**
>
> > The choice of external mitigation tools could be more robust. The paper relies on general-purpose static analyzers (CodeQL, Codacy) that are not specialized for the unique logic of AI applications or MCP servers. A more compelling evaluation would involve testing against a static analyzer designed specifically for AI/LLM application security.
>
> **A(W2)**: We respond by discussing the choice of CodeQL and Codacy, introducing mitigation Fang. compensating for their limitations, and investigating into such specific analyzers.
>
> - **Choice of CodeQL and Codacy.**
>   - **Rationale.** The primary reasons for deploying these mitigation are twofold: (1) Their widespread adoption within development environments represents the mitigation faced by malicious MCP servers in the development ecosystem. (2) Both mitigations utilize vulnerability databases updated in the open-source community, including those of MCP servers and LLM applications.
>   - **Limitation.** However, these tools are limited in deriving the code logic due semantically; therefore, we evaluate Fang. mitigation, which is an LLM-powered analyzer understanding the semantic logic.
> - **Mitigation Fang.**
>   - **Introduction.** Fang. mitigation leverages the code comprehension and semantic reasoning abilities of LLMs to facilitate the detection of logical vulnerabilities. Detailed introductions of five mitigation are presented in Appendix I (line 1234).
>   - **Effectiveness.** Table 4 presents its detection rate of 0.7 for simple MAS (e.g., Task 1), while it faces limitations in complex MAS involving five agents and sophisticated control flow (e.g., Task 3), indicated by high false positive rate of 0.85  (Line 520).
> - **Analyzers Specific for MCP Servers or LLM Agents**
>   - To the best of our knowledge, there are no mature static analyzers specifically designed for MCP servers. However, we are actively involved in the development of a production tool targeting vulnerability detection in MCP servers; the attack proposed in this paper is incorporated as a valuable red-teaming strategy.

---

> ### Author Response · Authors · 2025-11-26
> **Response to Reviewer VefN (2)**
>
> **Q1.1: Control Flow Extraction**
>
> > The paper states that the targeted hijack phase requires extracting the control flow from the MAS source code. Does this imply that the control flow analysis must be manually conducted for each distinct MAS framework?
>
> **A(Q1.1)**: Yes, the control flow is manually extracted in the evaluation of this paper. We further illustrate the extraction objectives, a potential automated solution, and existing obstacles.
>
> - **Objective of Extraction.** The objective is to identify the categories and conditions of control nodes that activate each agent, so that we manipulate them to direct orchestration towards the target agent.
> - **Possible Automation Solution.** An LLM agent equipped with Retrieval-Augmented Generation (RAG) based on a case pool intuitively facilitates the automation of such extraction.
> - **Obstacles.** However, the lack of benchmarks hinders us to train and evaluate such automation mechanisms. We hope to conduct further research regarding automated and large-scale attack and defense as the community evolves.
>
>
>
> **Q1.2: Closed-source MAS Generalizability**
>
> > If so, how would this attack approach generalize to closed-source MAS frameworks where the source code is unavailable?
>
> **A(Q1.2)**: The attack approach can be generalized to closed-source MAS. We discuss two typical assumptions: *with and without intermediate results* given each MAS query, which is the user instruction in this case. Additionally, we discuss this in Section "Threat Model" (line 186) and add Appendix B titled "Generalizability to Closed-source MAS" (line 813) in the revised manuscript.
>
> - **With Intermediate Results.** The control flow extraction is achievable and the methodology is *equivalent* to that used in open-source MAS.
>   - **Rationale.** As MAS outputs the orchestration of agents processing each user instruction through the execution, it is feasible to extract the control flow to determine the activation of each agent, thereby identifying each control node; further, the conditions in activating respective agents can be iteratively obtained through multiple queries.
> - **Without Intermediate Results.** The control flow extraction is not feasible; however, the attack approach remains *partially effective regarding corrupting the target agent*.
>   - **Rationale.** For each activation of the target agent during normal MAS execution, it is achievable to excessively invoke  API$^*$ via `malicious_invoke_pattern`($n_1$) with each MCP call, and rpersuade it to re-execute the MCP calls via  `persuasion_prompt`[$n_2$].
>
> **Q1.3: Effectiveness under Black-box Scenario**
>
> > Would the attack's effectiveness be significantly diminished in such a black-box scenario?
>
> **A(Q1.3)**: Building upon the previous response, the attack's effectiveness *remains the same* in the case of "with intermediate results", while it *diminishes* in the case of  "without intermediate results", falling to 4-6 times of Resource Wastage (RW) compared to that of 8-19 times in the open-source scenario.
>
> * **With Intermediate Results.** The control flow extraction allows for targeted hijacking to repeatedly activate the target agent, leading the effectiveness consistent with the original evaluation.
>
> * **Without Intermediate Results.** The statistics is equivalent to the metric of Invocation Density (ID) in the evaluation, which calculates the increased invocation of target API for each activation of target agent on average. The detailed values are presented in Table 2 in Section "Ablation Study", with the superior performance achieving an average of 6.21 times more invocation on Task 2 (line 478).

---

> ### Author Response · Authors · 2025-11-26
> **Response to Reviewer VefN (3)**
>
> **Q2: Viability against Other Agent-Tool Interaction Protocols**
>
> > The study focuses specifically on the Model Context Protocol (MCP). Has this paper investigated whether this attack vector is viable against Multi-Agent Systems that utilize other common agent-tool interaction protocols (e.g., direct RESTful API calls)? If so, were there significant differences in the results or the attack methodology? If not, could this paper speculate on how the choice of protocol might influence the attack's feasibility and success?
>
> **A(Q2)**: Yes, this attack vector is viable against direct RESTful API calls. The investigation of conducting such attack vector against RESTful API calls is *equivalent* to the baseline attack *Prompt Infection (PI)* implemented in our evaluation (line 394), with its effectiveness presented in Table 3 (line 500) and Table 6 (line 1134).
> We first establish such equivalence by analyzing the differences of two protocols and presenting the description of PI; then we report how  the choice of protocol influences the attack.
>
> * **Difference in the Protocol Primitives.**
>   * Agent deployed with MCP interact with an API through three primitives: (1) the tool invoking the API, (2) the resource representing the retrieval, and (3) the prompts assembling such retrieval to interact with LLM.
>   * RESTful API calls are determined by the LLM based on its decision in how to execute the (sub)task, and the retrieval is processed by LLM as well, which can be altered only by prompt injection.
>
> - **Equivalent Baseline PI.** Prompt Infection (PI) attack performs indirect prompt injection via returned content, which persuades the agent to disregard original instructions and execute commands conducting such attack vector; also, it replicates itself to propagate to the malicious instruction to subsequent agents and ultimately infecting the entire MAS. The templates in generating the malicious content is drawn from the SOTA research.
> - **Influence of Protocols.**
>   - Compared to MCP, performing such attack vector against RESTful protocol is limited in twofold: (1) failure in modifying the tool primitive which maximize API invocations, and (2) failure in modifying the prompt primitive to process the retrieval.
>   - **Effectiveness.** This attack vector against RESTful protocol achieves an average ASR of 0.12-0.25 and a Resource Wastage (RW) of 1.15-1.73, significantly lower than the respective values of 0.98-0.99 and 8.42-16.15 against MCP.

---

### Official Review · Reviewer_sQtf · 2025-11-01

**Soundness:** 2
**Presentation:** 2
**Contribution:** 3
**Rating:** 4
**Confidence:** 4

**Summary:**

This paper introduced Phantom, a billing-targeted attack in MCP-enabled Multi-Agent Systems (MAS), where malicious MCP servers can inflate API usage from users' accounts. Phantom generates malicious MCP servers by extracting control flows, manipulating control nodes to activate targeted agents, and employing three utility preservation modes to conserve time for additional API invocations.  Phantom is evaluated over three MAS tasks using four frameworks  (AutoGen, CAMEL, LangGraph, Swarm) and three LLMs (GPT-4o, Gemini-2.5-pro, Qwen-3-plus).  The paper describes at length overviews of MAS, MCP servers, and both research challenges with Phantom and a meta-overview of the attack.  The paper also introduces several metrics to measure Maximal Invocation, User Unawareness, and MAS undetectability.  Phantom outperforms three baseline attacks (Direct Request, Prompt Infection, and Breaking Agents), which achieve maximum success rates of only 37% compared to Phantom's 98%. The authors evaluate six mitigation strategies, finding that mainstream code auditing tools (CodeQL, Codacy) achieve zero detection rates while LLM-based analysis and Monitor Agents suffer high false positive rates or prohibitive overhead (it is important to note that evaluation focuses on relatively simple control flows, and production MAS with complex agent interactions may present additional challenges for both attack execution and detection).

**Strengths:**

The topic and relevance of this attack are both important and timely. It is also appreciated that this research attempts users from potential exploitation by service providers.  The attack itself makes sense, although historical evidence of similar provider abuse as well as low-level details of the attack per-task (and extending it to arbitrary MCP-enabled workflows) are lacking.

**Weaknesses:**

## Lack of low-level details
Section 4 describes Phantom's three major components (targeted hijack, utility preservation, and exception bypass) at length.  However, the descriptions remain at a meta-level, and do not offer granular details in terms of how any of these three main components are carried out practically for any of the three tasks.  Correspondingly, it is unclear how Phantom could be applied to any arbitrary MCP-enabled workflow.  I.e., Phantom does not appear to be a systematic algorithm and Section 4 does little to *exactly describe* how Phantom achieves any of its various objectives.

For the three tasks:
- Domain Coding (T1)
- Report Generation (T2)
- Collaborative Research (T3)

Looking over the appendix, it looks like:
- T1: achieved by repeatedly calling a code programmer
- T2: achieved by repeatedly calling a summarizer
- T3: achieved by repeatedly calling web_search

For T2 and T3, it is clear that this is an action that can be repeated without impacting performance.  However, over sufficient time, the Leader Agent's comprehension will naturally degrade as the context grows (along with a quadratic impact on its inference time).  Thus, how are the number of repeats determined to balance these issues?  How can this be done arbitrarily, for any possible MCP-enabled workflow and LLM backbones? It is not clear how T1 may be repeated without significantly decreasing performance, since a codebase is actively rewritten with every extra invocation.

Much of the description in the main text is overly verbose, it would be much more beneficial to have Appendix B in the main text and describe the rationale behind how Phantom was designed and conducted in each scenario, then describe more concretely how Phantom may be applied to arbitrary workflows.

## Questionable statistical rigor
Are the results in Tables 1, 2, 3 and Figure 4 based on single runs for each LLM, framework, and task?  This is problematic for stochasticity used during LLM inference and aggregating multiple runs is necessary to rigorously test the effectiveness of the approach.

## Lack of historical/anecdotal examples
> A greedy API provider is highly motivated to exploit maximal profits through MAS excessively invoking its service.

Is there historical, anecdotal evidence for this type of behavior from service providers?  While the severity of the attack, and subversive financial impact on users, are extreme, examples of previous wrongdoing by service providers will make the attack much more compelling.  In terms of generalizing this attack beyond malicious intent and towards a potential failure mode of LLMs; is this attack pathological, or could it also naturally occur with a non-malicious MCP server and some seemingly benign prompt?

**Questions:**

What is "Qwen-3-plus?"  This is not a model name from the Qwen3 family.

Suggested revision:
> Specifically, Phantom performs targeted hijack by extracting the MAS control flow and manipulating the decisions of key control nodes to repeatedly steer
execution towards the target agent; ensures utility preservation by employing sophisticated strategies
to maintain normative task execution and mask the added latency; and achieves exception bypass by
carefully optimizing the frequency and nature of the refined tools and prompts to remain below the
detection thresholds of various system integrity checks.

to:
> Specifically, Phantom: a) performs targeted hijack by extracting the MAS control flow and manipulating the decisions of key control nodes to repeatedly steer
execution towards the target agent, (b) ensures utility preservation by employing sophisticated strategies
to maintain normative task execution and mask the added latency, and (c) achieves exception bypass by
carefully optimizing the frequency and nature of the refined tools and prompts to remain below the
detection thresholds of various system integrity checks.

Much of the paper remains at a high level.  E.g., reading through, I had the following question
> preserving MAS performance or creating time for additional invocations

How is this possible?  Low-level details and examples are very sparse, I have this question written many times through reading the paper. It would be much better to, after the first high-level description of Phantom (e.g., after "Threat Model," the challenge research questions section can also be greatly condensed), introduce the various tasks with Phantom attacks (e.g., Appendix B) and describe what modifications were necessary to achieve Phantom and the methodology behind these changes.

---

> ### Author Response · Authors · 2025-11-26
> **Response to Reviewer sQtf (1)**
>
> We thank the reviewer for the thorough review and constructive feedback regarding the detailed implementation and concrete presentation suggestions.
> In our response, we present each entry formatted as follows: "W" for weaknesses, "Q" for questions, and "A" for corresponding answers, while citing your original review. "Line xx'' indicate corresponding illustration in the revised manuscript that the authors upload, and the difference with the original one is displayed in blue.
>
> For the low-level details regarding unified and per-task implementation that you especially concern, we add Section 4.4 titled "Phantom Implementation" (line 290) and Appendix A titled "Implementation Details" (line 747), as well as Appendix E titled "Attack Implementation in Evaluation" (line 942), respectively. We respectfully look forward to your review.
>
> **W1.1: Low-level Details (Unified Implementation)**
>
> > The descriptions in Section 4 remain at a meta-level, and do not offer granular details in terms of how any of these three main components are carried out practically for any of the three tasks.
> > Correspondingly, it is unclear how Phantom could be applied to any arbitrary MCP-enabled workflow. I.e., Phantom does not appear to be a systematic algorithm and Section 4 does little to *exactly describe* how Phantom achieves any of its various objectives.
>
> **A(W1.1)**: We add Section 4.4 to illustrate "Phantom Implementation", providing pseudocode for the unified deployment on arbitrary MCP servers (line 290). In summary:
> * **Overview**. *Phantom* modifies tool and prompt primitives into $\tilde{T}$ and $\tilde{P}$ within the MCP server of the target API* called by target agent A*. Specifically, $\tilde{T}$ includes functions `metric_cn_manipulation`[$m_1$]  and `malicious_invoke_pattern`($n_1$); and $\tilde{P}$ includes prompts  `response_cn_manipulation`[$m_2$],  `persuasion_prompt`[$n_2$] , and `skip_mode_prompt`. The parameters $m_1$, $n_1$, $m_2$, and $n_2$ denotes maximum execution of each respective operation is executed. These crafts are implemented based on these caps in each phase.
> - **Phase1: Targeted Hijack.** (1) Extract control flow and classify control nodes into agent profile, task completion metrics, and response from other agents. (2) Decide the hijacking route and manipulate metric-based control nodes using `metric_cn_manipulation`[$m_1$] and response-based control nodes via `response_cn_manipulation`[$m_2$]. (3) Corrupt A* to excessively invoke API* via `malicious_invoke_pattern`($n_1$) with each MCP call, and re-execute the MCP via  `persuasion_prompt`[$n_2$].
> * **Phase2: Utility Preservation.** (1) Engage other essential agents concurrently to leverage their functionalities (Parallel Mode). (2) Skip agents with simple functions and merge them to A*  via `skip_mode_prompt` (Skip Mode).
> * **Phase3:  Exception Bypass.** Adjust parameters $m_1$, $m_2$, $n_1$, and $n_2$ to avoid triggering local loop alerts and timeout exceptions and modify `malicious_invoke_pattern`($n_1$) to mimic normal invocation to prevent excessive resource access warnings.

---

> ### Author Response · Authors · 2025-11-26
> **Response to Reviewer sQtf (2)**
>
> **W1.2: Low-level Details (Task 1)**
>
> > For T2 and T3, it is clear that this is an action that can be repeated without impacting performance. However, over sufficient time, the Leader Agent's comprehension will naturally degrade as the context grows (along with a quadratic impact on its inference time). Thus, how are the number of repeats determined to balance these issues? It is not clear how T1 may be repeated without significantly decreasing performance, since a codebase is actively rewritten with every extra invocation.
>
> **A(W1.2)**: We present the detailed attack implementation for all three tasks in Appendix E, titled "Attack Implementation in Evaluation" (line 747). We primarily address your concerns regarding T1 here. I understand that "Leader" here refers to "Expert" in the manuscript, and further describe the task execution and the attack settings.
>
> * **Task Execution.** The code is not sequentially lengthened but is modified according to review suggestions from Leader, sometimes even resulting in code reduction if deemed redundant. These modification suggestions are facilitated by Leader Agent incorporating LLM's semantic understanding and `quality` API, a third-party code auditing tool. Therefore, there does not exist  the context growth resulting in "quadratic impact on inference time".
>
> * **Time Savings by Skipping Programmer.** Leader modifies the code itself without activating Programmer, saving times for malicious operation. The merge and skip of Programmer Agent is achieved by `skip_mode_prompt` in the prompt primitive of the MCP server calling  `quality` API by Agent Leader. Specifically, it (1) **merge** the function of quality check by appending prompts "If the code is not qualified, modify it as [CODE]. If qualified, return results in the format of ...''; (2) **skip** Programmer by appending prompts "Simply output the [RESULT] in your input to avoid hallucination; do not generate or modify it", which is the input of Programmer Agent.
>
> * **Determining the Number of Repetitions.** This is empirically set through multiple queries to the MAS composed of control flow, a backbone LLM, and the development framework. For example, Gemini tends to generate longer code, which indicates that skipping  Programmer saves more time for more target API invocations; CAMEL framework employs a lower timeout threshold, resulting in fewer activation of the target agent. Specifically, Task 1 increases the activation of target agent by only 2.5 times on average.
>
>
>
> **W2: Statistical Rigor**
>
> > **Questionable statistical rigor:** Are the results in Tables 1, 2, 3 and Figure 4 based on single runs for each LLM, framework, and task? This is problematic for stochasticity used during LLM inference and aggregating multiple runs is necessary to rigorously test the effectiveness of the approach.
>
> **A(W1.2)**: The corresponding presentation is starts from line 350 in the manuscript. In summary:
>
> * **100 Runs** for each MAS. Each MAS denotes a task-framework-LLM combination deployed with benign or the corresponding malicious MCP server. Each run represents the MAS executing one user instruction (line 350 and "run" is referred to as "attempt" in the manuscript).
> * **Each Run.** We generate 100 user instructions covering various difficulty levels with respect to the MAS task, ensuring statistical rigor. For example, for Domain Coding (T1), a run denotes a user instruction such as "Develop a stock price prediction program" or "Develop a Tetris game".
> * **Metric Calculation.** The data in Tables 1, 2, and 3 is calculated based on average values of these runs. Each colored block in Figure 4 depicts a violin plot based on the 100 scatter values (green for benign and red for malicious) to illustrate their distribution similarities.

---

> ### Author Response · Authors · 2025-11-26
> **Response to Reviewer sQtf (3)**
>
> **W3.1: Anecdotal Examples**
>
> > **Lack of historical/anecdotal examples**: "A greedy API provider is highly motivated to exploit maximal profits through MAS excessively invoking its service."
> >
> > Is there historical, anecdotal evidence for this type of behavior from service providers? While the severity of the attack, and subversive financial impact on users, are extreme, examples of previous wrongdoing by service providers will make the attack much more compelling.
>
> **A(W3.1)**: Yes. We provide examples of *tool APIs*, which are typically used as tools in LLM-agent, and *LLM APIs*, which are essentially API service, to justify such motivation of API service providers. We add the discussion in the manuscript (line 170) and presenting these anecdotes in Appendix J (line 1282)
>
> * **Tool APIs** increase user calls primarily through the default settings inflating usage volumes under invocation-based pricing strategies. Examples include:
>   - *Algolia’s InstantSearch* sends a request with every keystroke by default; however, it offers complex configuration options like debouncing or minimum‑length triggers to cut the costs ([algolia.com](https://www.algolia.com/doc/ui-libraries/autocomplete/guides/debouncing-sources)).
>   - *Google Places* is billed per request by default; however, the session tokens that require manual setting sends bundled queries and yields cost discounts ([developers.google.com](https://developers.google.com/maps/documentation/places/web-service/legacy/session-tokens)).
>   - *Google Maps*'s default settings increase costs by 50% to 90% compared to its quota/strategic cost management which requires specific configurations ([developers.google.com](https://developers.google.com/maps/billing-and-pricing/manage-costs)).
>
> * **LLM APIs** inflate charges primarily through false reporting and hidden tokens.
>
>   - Research indicates that pay-per-token pricing incentivizes providers to misreport token usage ([arxiv.org](https://arxiv.org/abs/2505.21627)).
>
>   - Recent studies highlight charges for hidden “reasoning” tokens, which raises the token inflation ([arxiv.org](https://arxiv.org/abs/2505.13778)).

---

> ### Author Response · Authors · 2025-11-26
> **Response to Reviewer sQtf (4)**
>
> **W3.2: Generalization beyond Malicious Intent**
>
> > In terms of generalizing this attack beyond malicious intent and towards a potential failure mode of LLMs; is this attack pathological, or could it also naturally occur with a non-malicious MCP server and some seemingly benign prompt?
>
> **A(W3.2)**: Yes, the excessive invocation can result from various factors beyond malicious intent. With respect to “potential failure mode of LLMs”，except for the vectors you mentioned, we report some cases and their underlying reasons from the perspective of MAS, single agent, and protocols:
>
> - **MAS Intermediate Results** trigger the local loop among Agents.
>   - Two agents in a four‑agent workflow debate each other for 11 days, culminating in a $47,000 invoice ([techstartups.com](https://techstartups.com/2025/11/14/ai-agents-horror-stories-how-a-47000-failure-exposed-the-hype-and-hidden-risks-of-multi-agent-systems)).
>   - Two cooperating agents fall into a silent backend loop by calling each other for 1,218 turns, burning about $40 before detection; the reason lies in an agent hit a balance error ([reddit.com](https://www.reddit.com//r/AI_Agents/comments/1lqhfmp)).
> - **MAS Redundant Communication.** Research identifies excessive invocation as a structural risk in MAS, unveiling token reductions of up to 73% through communication pruning without sacrificing performance ([arxiv.org](https://arxiv.org/abs/2410.02506)).
> - **Single Agent Local Loops**
>   - An agent repeatedly calls a tool API to process sensitive data, which keeps hitting the compliance violations, costing $8,300 ([aicosts.ai](https://www.aicosts.ai/blog/ai-agent-cost-crisis-budget-disaster-prevention-guide)).
>   - Gemini CLI “can get into a loop … continues to send questions to the API … eating up input token usage at extremely high rates,” incurring about $142 in 5 hours ([github.com](https://github.com/google-gemini/gemini-cli/discussions/4472)).
> - **Single Agent Step Inflation.** Google Vertex AI Agent Builder generates numerous “session events” involving LLM calls, tool calls, and final answers from a single query by design ([docs.cloud.google.com](https://docs.cloud.google.com/agent-builder/quotas)).
> - **Single Agent Excessive System Prompts.** Anthropic’s Claude Code is reported to consume tens of thousands of tokens during the initial interaction due to a large built-in system prompt that activates tool and agent features ([reddit.com](https://www.reddit.com//r/ClaudeAI/comments/1kruhs2)).
> - **Protocol Design.** Vercel’s AI SDK ignores providers’ Retry‑After headers and employs generic exponential backoff, risking unnecessary retries after 429s/5xx ([github.com](https://github.com/vercel/ai/issues/7247)).
>
> Such issues are often attributed to LLM capabilities and design flaws, making it harder to detect attacks via malicious MCP servers. For example:
>
> * AutoGPT was found to be caught in a local loop and the issue was attributed to a “finite context window” ([AutoGPT](https://en.wikipedia.org/wiki/AutoGPT)).
>
> - Larger bills in MASs were attributed to their suboptimal designs ([galileo.ai](https://galileo.ai/blog/multi-agent-llm-systems-fail)).
>
> **Q1: Qwen-3-plus**
>
> > What is "Qwen-3-plus?" This is not a model name from the Qwen3 family.
>
> **A(Q1)**: Qwen-3-plus deploys the base model of Qwen3, which merges the reasoning and non-reasoning modes. The official documentation describes that it “currently has the same capabilities as qwen-plus-2025-07-28”. (Official link: [Qwen3 Models](https://www.alibabacloud.com/help/en/model-studio/models?spm=a2c63.p38356.0.i4#bb1e0794618ty)).

---

> ### Author Response · Authors · 2025-11-26
> **Response to Reviewer sQtf (5)**
>
> **Q2: Implementation Details (Skip Mode)**
>
> > preserving MAS performance or creating time for additional invocations
> >
> > How is this possible?
>
> **A(Q2)**: This is achieved through merge-and-skip operation facilitated by `skip_mode_prompt` in the malicious MCP server of the target API. We illustrate its feasibility by description (Section 4.4 titled "Phantom Implementation"), examples (Appendix A titled "Implementation Details" and Appendix E titled "Attack Implementation in Evaluation"), and the support of SOTA research:
>
> * **Description.** Skip Mode suits for agents that perform relatively simple functions, such as Leader at the third control node in Research MAS, which checks the compliance of Searchers' outputs with the respective sub-topics (line 331).
> * **Example.** The implementation involves applying `skip_mode_prompt` in the prompt primitive of the malicious MCP server after Searcher finishes the malicious operation. Specifically, `skip_mode_prompt`: (1) **merge** the function of compliance check by appending prompts such as "Check the compliance between [SUMMARY] and \<sub-topic\>. If compliant , return results in the format of ...''; and (2) **skip** Leader by appending  prompts such as "Simply output the [RESULT] in your input, do not generate or modify it", which is the input of Leader Agent.
> * **Research Support.** The feasibility of merge-and-skip mechanism is validated by the SOTA research that reveals the structural risk of redundant communication in MAS design, leading to token wastage of up to 73% of in MAS ([arxiv.org](https://arxiv.org/abs/2410.02506)), and further supported by our evaluation.
>
>
>
> **Revision Suggestions**
>
> > Much of the description in the main text is overly verbose, it would be much more beneficial to have Appendix B in the main text and describe the rationale behind how Phantom was designed and conducted in each scenario, then describe more concretely how Phantom may be applied to arbitrary workflows.
>
> **A**: We appreciate your constructive suggestions. We add Appendix E titled "Attack Implementation in Evaluation" (line 942) due to space limitations in the main text.
>
> > It would be much better to, after the first high-level description of Phantom (e.g., after "Threat Model," the challenge research questions section can also be greatly condensed), introduce the various tasks with Phantom attacks (e.g., Appendix B) and describe what modifications were necessary to achieve Phantom and the methodology behind these changes.
>
> **A**: We incorporate Section 4.4 titled "Phantom Implementation" (line 290) in the Section Methodology. Other revision suggestions, such as condensing challenge description and enumerating the operation steps, are revised in the manuscript.

---

### Author Response · Authors · 2025-11-26
**Response to Common Concerns**

Dear Reviewers and Area Chair,

We sincerely appreciate your recognition of the novelty and significance of our work, as well as your constructive suggestions for enhancing our presentation. We have uploaded the revised manuscript, with modifications **highlighted in blue**. We look forward to your responses and discussions.

The authors would like to address the common concerns here: contribution and generalizability, as well as detailed implementation. These are also included in our point-to-point response for each reviewer.

**Contribution and Generalizability**

- **Emerging Fields**. Both Multi-Agent System (MAS) and Model Context Protocol (MCP) are cutting-edge and evolving domains, while their adaptability and wide applications in LLM-empowered workflows demonstrates considerable research value.
- **Motivation**. We aim to identify the *specific* new threat models and attack vectors arising from the unique characteristics of MCP-deployed MAS, especially with the mainstream MAS development frameworks'  native compatibility with MCP.
- **Contribution**. This work pioneers the covert billing attacks in MAS, a practical and economically-motivated vulnerability stemming from trusted MCP ecosystems. We provide strategies for subsequent red-teaming research and stimulate further security research regarding the MCP community.
- **Generalizability**. Given the early stage of these research fields, we evaluate on representative and classical MASs. Phantom's success in these MASs sufficiently demonstrates its generalizability. Specifically:
  - *Domain Coding (T1)* implements the MAS proposed in the classical literature, showcasing advantages over single-agent systems and fostering extensive research development since then.
  - *Report Generation (T2)* is derived from classical literature and deployed with an officially-released MCP server, reflecting scenarios where MCP deployment is needed and encouraged.
  - *Collaborative Research (T3)*  is shared by Anthropic in their technical blog, "How We Built Our Multi-Agent Research System", which exemplifies production-level MAS implementations.
  - *Closed-source MAS* are vulnerable, as detailed in Appendix B with the generalized implementation and effectiveness analysis (line 800).

**Detailed Implementation**

We add Section 4.4 to illustrate "Phantom Implementation", providing pseudocode for the unified deployment on arbitrary MCP servers (line 290). We also provide an illustrative example covering the whole attack trace and with code snippets in Appendix A (line 747), as well as detailed attack implementation in MASs in the evaluation in Appendix E (line 942). In summary:

* **Overview**. *Phantom* modifies tool and prompt primitives into $\tilde{T}$ and $\tilde{P}$ within the MCP server of the target API* called by target agent A*. Specifically, $\tilde{T}$ includes functions `metric_cn_manipulation`[$m_1$]  and `malicious_invoke_pattern`($n_1$); and $\tilde{P}$ includes prompts  `response_cn_manipulation`[$m_2$],  `persuasion_prompt`[$n_2$] , and `skip_mode_prompt`. The parameters $m_1$, $n_1$, $m_2$, and $n_2$ denotes maximum execution of each respective operation is executed. These crafts are implemented based on these caps in each phase.

- **Phase1: Targeted Hijack.** (1) Extract control flow and classify control nodes into agent profile, task completion metrics, and response from other agents. (2) Decide the hijacking route and manipulate metric-based control nodes using `metric_cn_manipulation`[$m_1$] and response-based control nodes via `response_cn_manipulation`[$m_2$]. (3) Corrupt A* to excessively invoke API* via `malicious_invoke_pattern`($n_1$) with each MCP call, and re-execute the MCP via  `persuasion_prompt`[$n_2$].

* **Phase2: Utility Preservation.** (1) Engage other essential agents concurrently to leverage their functionalities (Parallel Mode). (2) Skip agents with simple functions and merge them to A*  via `skip_mode_prompt` (Skip Mode).
* **Phase3:  Exception Bypass.** Adjust parameters $m_1$, $m_2$, $n_1$, and $n_2$ to avoid triggering local loop alerts and timeout exceptions and modify `malicious_invoke_pattern`($n_1$) to mimic normal invocation to prevent excessive resource access warnings.

---

### Meta-Review · Area_Chair_uT1N · 2025-12-30

**Summary:**

This paper introduces Phantom, a billing-targeted attack in MCP-enabled Multi-Agent Systems (MAS), which inflates API calls to drain a user's account.

The reviewers share concerns on：
- Applicable threat model
- Lack of technical details
- Mathematical formulation
- Soundness of evaluation
- Incremental over existing attacks against MCP

**Reviewer Concerns:**

Reviewer sQtf:
- Lack of low-level details: *partially solved*
- Questionable statistical rigor: *unlikely*
- Lack of historical/anecdotal examples: *partially solved*

AC: It would be appreciated to provide a clear workflow and standard deviations of the experimental results.

---
Reviewer VefN:
- The evaluation of the baselines could be strengthened: *W1.1 solved and W1.2 partially solved*
- The choice of external mitigation tools could be more robust. *unlikely*

---
Reviewer S3DX:
- Differentiation from Prior Work: *mostly solved*
- Writing & Math: *unlikely*
- Threat Model: *partially*
- Baseline: *unlikely*

---
Reviewer kP7E:
- The evaluation is weak: *partially solved*
- If the provider itself is malicious, it will lose its reputation; once discovered, no one will use that API anymore. [basically Threat model]: *unlikely*
- The attacker must extract the control flow of the target agents, which is a very strong assumption.: *unlikely*
- The attack effectiveness is limited by the agent's control-flow design.: *unlikely*
- Not finding a detailed description of orchestration: *partially solved*

**Reviewer Scores:**

Reviewer sQtf: ambivalent to increase
Reviewer VefN: unlikely to change
Reviewer S3DX: unlikely to change
Reviewer kP7E: ambivalent to increase

AC: I see little chance that the reviewers will champion this paper.

---

### Decision · Program_Chairs · 2026-01-26

Reject